# Temperate Lianas Have More Acquisitive Strategies than Host Trees in Leaf and Stem Traits, but Not Root Traits

**DOI:** 10.3390/plants11243543

**Published:** 2022-12-15

**Authors:** Zhe Zhou, Binzhou Chen, Hongru Zhao, Junjie Yi, Shiqiang Liu, Dan Tie, Jinshi Xu, Shu Hu, Yaoxin Guo, Ming Yue

**Affiliations:** Key Laboratory of Resource Biology and Biotechnology in Western China (Ministry of Education), Northwest University, Xi’an 710069, China

**Keywords:** functional traits, lianas, host trees, resource acquisition strategies, temperate forest

## Abstract

Increasingly, tropical studies based on aboveground traits have suggested that lianas have a more acquisitive strategy than trees, thereby possibly explaining the increase in lianas relative to trees in many tropical forests under global change. However, few studies have tested whether this pattern can be extended to root traits and temperate forests. In this study, we sampled 61 temperate liana-host tree pairs and quantified 11 commonly studied functional traits representative of plant economics in roots, stems, and leaves; we aimed to determine whether root, stem and leaf traits are coordinated across lifeforms, and whether temperate lianas are also characterized by more fast and acquisitive traits than trees. Our results showed that leaf and stem traits were coordinated across lifeforms but not with root traits, suggesting that aboveground plant economics is not always correlated with belowground economics, and leaf and stem economic spectra cannot be expanded to the root directly. Compared with host trees, lianas had more acquisitive leaf and stem traits, such as higher specific leaf area and lower leaf dry matter content, leaf carbon content, leaf mass per area, and wood density, suggesting that lianas have a more acquisitive strategy than host trees in the temperate forest. The differences between lianas and trees in plant strategy may drive their contrasting responses to the changing temperate forest environment under global change.

## 1. Introduction

Global change (elevated CO_2_, drought, habitat fragmentation, etc.) is altering the vegetation composition and structure of many forest ecosystems, and further affecting the capacity of these forests to act as a carbon sink [1]. One major phenomenon in the past decade is that the abundance of lianas has been changing disproportionately with trees as a result of increasing drought and forest fragmentation. For example, reports have increasingly suggested that lianas are growing in density and biomass, relative to trees in tropical forests [2,3,4,5,6]. The expansion of lianas is bound to affect the structure and function of forest ecosystems. Therefore, revealing the specific mechanisms favoring lianas and how lianas differ from coexisting tree species in resource acquisition and utilization strategies has become crucial to improve our ecological understanding of the phenomenon of liana expansion [2,7,8].

Plant functional traits are any morphological, physiological, or phenological feature that affects plant growth, survival, and reproduction, reflecting long-term adaptations to environmental conditions [9]. Therefore, how lianas and trees differ in resource acquisition and utilization strategies likely depends on their functional traits [10]. On the basis of functional traits, recent studies in tropical forest areas explored how lianas differ from trees [11,12]. Most of these studies found that lianas have more acquisitive characteristics, such as higher specific leaf area (SLA) and lower tissue density than trees [13,14], which allow lianas to benefit from high-light and nutrient-rich soils and thus perform better in dry and fragmented environments [12]. However, not all studies found that lianas have more acquisitive traits than trees. Several studies have indicated that lianas and trees do not differ in wood density (WD) [13] and leaf nutrient concentrations (nitrogen and phosphorus) [8]. In addition, the available literature is mostly focused on aboveground traits, and comparison between lianas and trees in root traits is lacking. Traits at the leaf, stem and root levels are expected to be coordinated across lifeforms because organisms function as an integrated whole unit [15]. Therefore, lianas are expected to be characterized by resource acquisitive root traits, such as high specific root length (SRL) and high nitrogen concentrations. For example, Liu et al. [16] found that plant leaf and root traits covaried across multiple spatial scales in Chinese semi-arid and arid ecosystems. However, the coordination between leaf, stem and root traits is seldom tested across lifeforms [12], and it is also unknown whether the difference in aboveground traits between lianas and trees can extend to root traits [8]. Therefore, more extensive studies including leaf, stem and root traits are needed to improve our understanding of the coordination of different organ traits and the difference between lianas and trees in ecology strategy.

Compared with those in the tropics, lianas in temperate forests are typically less abundant and diverse. However, the liana abundance in temperate forest areas is highest in disturbed areas and along forest edges [17,18]. A few previous studies have suggested that the expansion of lianas is occurring in these temperate forests under global change [10]. However, no direct study has focused on how lianas differ from coexisting tree species in plant strategies or has tested whether the pattern found in tropical forests can extend to temperate forests. Therefore, we have conducted a comparative study of lianas and host trees in a temperate forest of China on the basis of above and belowground functional traits, to examine how the resource acquisition strategies of temperate lianas differ from coexisting trees. Based on our current knowledge, we advanced these hypotheses: 1) root traits coordinate with leaf and stem traits in reflecting the resource strategies of temperate lianas and trees, and 2) like tropical forests, temperate lianas are characterized by more acquisitive traits than trees.

## 2. Materials and Methods

### 2.1. Study Area and Species Selection

The study site was located in Taibai Mountain (Mt. Taibai) of Qinling Mountains (33°49′ N–34°10′ N, 107°19′ E–107°58′ E), Shaanxi, China. The Qinling Mountains run east–west in central China and form a transitional zone between northern subtropical and warm temperate zones, thereby making this location a global biodiversity hotspot [18,19]. Mt. Taibai is the highest mountain of the Qinling Mountains, and natural vegetation types include *Quercus* forest (<2000 m), hardwood–conifer mixed forest (2000–2800 m), *Abies* forest (2800–3200 m), *Laris* forest (3000–3400 m), and alpine scrubs (>3400 m) along the altitudinal gradient [18]. Our sampling was conducted in the *Quercus* forests at an elevation between 1000 and 2000 m. *Quercus* forest had been disturbed for a long time, due to the lower altitude, before the reserve was established in the 1960s; therefore, most of the stands are secondary forests in different succession stages, and are where climbing plants are more abundant than other forest types [20]. This region has a warm temperate climate. Its mean annual temperature is 4 °C–10 °C, and the mean annual rainfall is 750–1000 mm, based on the meteorological data from 2000 to 2020.

To maintain macro and microenvironmental consistency, we selected lianas and their host trees as target species along line transects with 100 m altitudinal intervals. We defined a suitable liana for sampling as an individual that climbed to the canopy and had a definite host tree species. Finally, 61 liana-host tree pairs, including 5 liana species and 12 host tree species, were selected and sampled in a total of ten line transects between June and September 2020. (Appendix A).

### 2.2. Plant Trait Measurement

In the measurement of functional traits, 11 key traits concerning plant resource acquisition and utilization were determined (Table 1). These traits included leaf area (LA), SLA, leaf thickness (L_th_), leaf dry matter content (LDMC), mass-based leaf carbon and nitrogen concentrations (LCC and LNC, respectively), stem wood density (WD), mass-based root carbon and nitrogen concentrations (RCC and RNC, respectively), and specific root length (SRL). Trait measurements were conducted for each individual liana and tree.

In the measurement of leaf traits, 5–10 replicate canopy leaves per tree or liana were collected using carbon picking rods (20 m). Leaves were scanned and processed using the Motic Image Plus 2.0 to calculate LA. L_th_ was measured using a Vernier caliper at the same position of each leaf and avoiding the main and secondary veins. Following these measurements, the leaf fresh weight was measured, and all leaves were then dried at 80 °C for 72 h and reweighed to determine dry weights. We then measured the SLA, LMA, and LDMC using the equations: SLA = fresh LA (cm^2^)/leaf dry weight (g), LMA = leaf dry weight (mg)/fresh LA (cm^2^), and LDMC = leaf dry weight (mg)/fresh weight (g), respectively. LNC and LCC were measured using the EA3000 Elemental Analyzer (Italy).

For the measurement of stem woody density, a wood core from the bark inward to the center of the stem, per liana and tree, was extracted using an increment borer at breast height. In view of the wide difference in stem diameter between lianas and trees, we selected large-diameter corers (12 mm) for trees and small-diameter corers (5 mm) for lianas. WD was calculated using the equation: WD = mass (g)/volume (cm^3^) of the cylindrical core. The volume of a cylindrical sample was determined by measuring the length (L) and diameter (D) of the cylindrical core using the equation: volume of a cylindrical sample = (0.5D)^2^ × π × L [21]. All cylindrical cores were then dried to constant weight.

For root collection and trait measurement, we dug a certain number of fine roots (diameter ≤ 2 mm) for each liana and tree individual. Excavation was carried out without destroying the forest soil environment. After excavation, roots were transported to the laboratory immediately. Subsequently, the fine roots retrieved were rinsed with distilled water and cleaned with a fine brush. We then measured SRL, RCC and RNC. SRL is the ratio of fine root length to dry weight of fine root.

### 2.3. Data Analyses

We used generalized linear mixed effects models to evaluate root, stem, and leaf trait differences between lianas and trees. Life form (tree or liana) was a fixed effect, and family was a random effect. PCA was performed to determine the nature and number of the relevant axes of functional differentiation between lianas and trees on the basis of the correlation matrix of trait variables, further visualize the distribution of two lifeforms in the two-dimensional trait space, and test for overall associations between traits and the difference in ecological strategies between lianas and trees. For the PCA, we standardized each functional trait by transforming the normalized trait values into z-scores. The distribution of each life form in the two-dimensional space was represented by 95% “concentration” ellipses.

In addition, we compared the bivariate trait associations between lianas and trees. First, we estimated the all-species bivariate trait associations between functional traits by using the standard Pearson correlation analyses, which could reflect the proportion of variation in one variable that was accounted for by the variation in the other variable [22]. For significant all-species bivariate trait associations, we further tested whether these associations were consistent between lianas and trees by using the standard major axis analyses (SMA). In detail, we first tested the differences in the slopes of liana and tree regression lines. If slopes were not significant, we then tested the shifts in elevation (i.e., intercept) and whether growth forms were separated with a common slope.

## 3. Results

### 3.1. Multivariate Analyses and Trait Associations

In our study, we observed that leaf traits covaried with stem traits across lifeforms but were largely independent from the variation in root traits. In the PCA that included lianas and host trees, among the traits measured, the first two principal components accounted for 53% of the variance (Table 2). The first component was associated with leaf and stem traits. LDMC was the main contributor to the first component, followed by WD, SLA, LCC, and LMA. The first component ran from species with cheap and acquisitive leaves and stem (high SLA and low LDMC, LMA, LCC, and WD) to conservative leaves and stem (low SLA and high LDMC, LMA, LCC, and WD). The second component, which was independent from the first component, ran from species characterized by low RNC and RCC to species characterized by high RNC and RCC.

PCA showed contrasting differences between lianas and host trees. The first component scores of lianas and trees were significantly different (Appendix A), with lianas occupying the acquisitive side of the gradient and trees occupying the conservative side (Figure 1). Lianas had a significantly higher SLA and lower LDMC, LMA, LCC, and WD than trees (Figure 2). However, we did not observe differences between lianas and trees in the second component scores (Figure 1 and Appendix A). Lianas did not significantly differ in terms of SRL, RNC, and RCC (Figure 2).

### 3.2. Bivariate Trait Associations

Among all species, L_th_ was negatively associated with LA, LDMC, LCC, WD, and SRL (Table 3). LA was positively associated with LDMC, LNC, LCC, and WD (Table 3). LDMC was positively associated with LMA, LNC, LCC, and WD and negatively associated with SLA (Table 3). LMA was positively associated with LNC, LCC, and WD but negatively associated with SLA (Table 3). SLA was negatively associated with LNC, LCC, and WD (Table 3). LNC was positively associated with LCC and WD and negatively associated with RNC (Table 3). LCC was positively associated with WD (Table 3). RNC was positively associated with RCC (Table 3). Among the bivariate trait associations, the relationships between LDMC and SLA, LMA, and WD were strongest. About 60% of the variation in SLA and LMA and 73% of the variation in WD accounted for the variation in LDMC (Table 3).

For the significant all-species bivariate trait associations, lianas were not always consistent with host trees. The slope between LDMC and SLA for host trees was significantly steeper than that for lianas (Figure 3, Appendix A), indicating that the SLA of trees was sensitive to the changing LDMC. By contrast, the slope between LNC and WD for lianas was significantly steeper than that for host trees (Figure 3, Appendix A), indicating that the WD of lianas was sensitive to the changing LNC. In addition, the differences in the elevation of bivariate relationships showed that lianas exhibited higher LNC than host trees at any given LA and LCC (Figure 3, Appendix A). These observations were consistent with the data observed in the multivariate analyses, with lianas being more acquisitive than host trees.

## 4. Discussion

### 4.1. Coordination among Leaf, Stem, and Root Traits

Our data found a strong coordination between leaf and stem traits across lifeforms, but these aboveground traits do not covary with root traits. This observation rejected the first hypothesis, that root traits coordinate with leaf and stem traits in reflecting the resource strategies across lifeforms. First, the first axes of the multivariate PCA are associated with leaf and stem traits but do not include root traits, whereas the second axes, which are independent from the first axes, are mostly associated with root traits. Second, significant differences are observed between lianas and host trees in leaf and stem traits but not in root traits, suggesting that root traits are inconsistent with leaf and stem traits in reflecting the resource acquisition strategies of plants. Third, bivariate trait analysis shows significant associations between leaf and stem traits but not between aboveground and root traits.

A plant is generally considered as a whole with a set of coordinated traits from different organs responding to the environment [23]. Therefore, the “leaf economics spectrum” proposed by ecologists may exist for stems and roots [15,24]. Many studies found that leaf economics is correlated with stem economics in reflecting the ecological strategy of plant species. For example, species with high growth rates develop cheap leaf and stem, whereas species with low growth rates have the opposite suite of leaf and stem traits [19,25]. In agreement with those of previous studies, our results across lianas and trees further confirmed the coordination between leaf and stem traits across lifeforms. However, a decoupling between aboveground and root traits in our study suggested that root economics varies independently from leaf and stem economics. Similar results are observed in the study of Medeiros et al. [26], which found that leaf and root traits are not coordinated across environments for the plants of the genus *Rhododendron.* Despite the whole plant integration across all organs, at least three reasons are contributory to the independence of root traits from leaf and stem traits. First, plant economics hinges on the lifespan of the organ, which determines how long the organ must remain useful to return the investment in growing it [27]. However, the lifespans of leaf and stem may not be tightly correlated with roots [28,29]. Second, although plant roots have uptake functions like leaves, above and belowground resources and environments are apparently different. Aboveground resources, such as light, are characterized by temporal heterogeneity, particularly for canopy species; in contrast resources in the soil are spatially heterogeneous [30], potentially making above and belowground organs be characterized by different traits when exploring different resource and environment heterogeneities. In addition, mycorrhizae affect the dependence of plants on roots in belowground resource acquisition, and different plants generally have different mycorrhizal colonization [8,31]. These reasons may cause root traits to not always mirror leaf and stem traits in reflecting plant ecology strategy.

### 4.2. Resource Acquisition Strategies in Lianas and Host Trees

Our data supported the second hypothesis that temperate lianas are characterized by more acquisitive strategy than host trees, confirming that the pattern found in tropical lianas and trees can extend to temperate forests. In the present temperate forest, lianas and trees predominantly differ in the main axis of the PCA, and lianas have higher SLA and lower LDMC, LCC, LMA, and WD than host trees. High acquisitive leaf and stem traits indicate that liana has a high photosynthetic rate and a high stem water transport efficiency at a low expenditure of structural carbon, thereby allowing lianas to adapt better to high light and dry conditions than trees. These results are consistent with the findings of many previous studies in tropical and subtropical forests, observing that lianas are faster at acquiring resources and less conservative than trees [3,4,7,8,11,32,33]. The exception is the LNC, with lianas having lower LNC than trees in the present forest. However, the difference in the SMA elevation between lianas and trees indicates that lianas have higher LNC than host trees at any given LA, suggesting that low LNC does not limit the leaf photosynthetic rate of lianas. In addition, the low LCC in lianas suggests labile liana leaf litter, which may stimulate nutrient cycling in soils and promote their growth [34,35].

In addition, we found that lianas have smaller and thicker leaves than host trees. Previous studies in tropical forests also observed that lianas have lower LA than trees, but these studies did not find a significant difference in L_th_ [8,12,13]. This result suggested that the small and thick leaves of lianas in our study may reflect the adaptation of these lianas to regional climate. Although acquisitive species are assumed to have large and thin leaves to maximize the interception of light and minimize carbon investment [36,37], leaf shape is plasticly and easily regulated by abiotic environments [38], and LA and L_th_ do not always correlate with other leaf traits [8,39]. In our study, the relatively small and thick leaves of lianas can be attributed to the adaptation of lianas to the canopy environment of a temperate forest. In the present temperate forest, the leaves of lianas residing in the upper canopy of the forest can intercept light and experience wider daily temperature ranges and larger vapor pressure deficits than those of host trees. In this case, small and thick leaves can maintain the optimal leaf temperature and efficient use of water for photosynthesis, and help lianas adapt to the exposed habitat [26,29,40].

### 4.3. Dynamic Prediction of Temperate Lianas in the Context of Global Change

Increasing studies over the past decade have suggested that global change, such as increasing drought and disturbance, is altering the relative proportion of lianas and trees in tropical forests, resulting in increased liana densities relative to trees [2,4,5]. Although we have evidence that lianas are increasing in many tropical forests, evidence regarding the changes in temperate lianas is lacking. Similar to tropical areas, temperate forests are experiencing multiple facets of global change, including rising atmospheric CO_2_ concentrations, climate warming, widespread forest fragmentation, and frequent drought [41,42]. Although we have little direct data on the changes in temperate lianas under global change, the increase in lianas relative to trees is probably occurring in many temperate forests due to the fast and acquisitive strategy of lianas found in the present temperate forest. Several previous studies in temperate forests indicated that disturbed and exposed forest environments can increase the abundance and diversity of lianas [17,18]; these studies suggest that the changing forest environment under global change may favor temperate lianas more than symbiotic trees, resulting in the abundance of lianas increasing relative to trees in temperate forests. Thus, in the future, long-term observation data are urgently needed to capture liana dynamics in temperate forests, because the increase in lianas relative to trees further changes the way that temperate forests uptake, cycle, and store carbon. In addition, although climbers have a similar growth form, they have evolved multiple methods of climbing. The previous studies have suggested that different climbing mechanisms have adapted to different forest environments. For example, DeWalt et al. [34] found that the relative abundance of stem twiners in a tropical rain forest increase with forest stand age, while the relative abundance of tendril climbers decreased with forest stand age. It suggests that different lianas and their climbing mechanisms may respond differently to global climate change and forest disturbance. Therefore, in future, testing the ecological strategies among the different climbing mechanisms is necessary to deeply understand and predict the dynamic of this life form under global change.

## 5. Conclusions

We observed that leaf traits coordinated with stem traits across life forms but did not covary with root traits in the present temperate forest, suggesting that aboveground plant economics was not always correlated with belowground economics. Compared with host trees, lianas were characterized by more acquisitive leaf and stem traits in the present temperate forest, and this result was consistent with the findings in many tropical forests, observing that lianas had a more rapid resource acquisition strategy than trees. The differences between lianas and trees in plant strategy may drive the contrasting responses to the changing temperate forest environment under global change and call attention to the dynamics of temperate lianas. 

## Figures and Tables

**Figure 1 plants-11-03543-f001:**
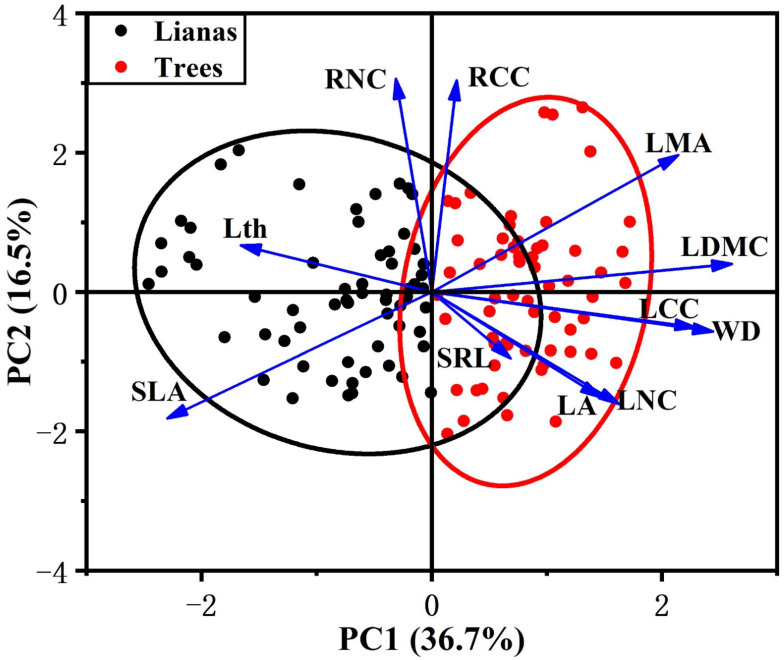
Distribution of lianas and host trees in traits space defined by the first two PCA axes. Black represents lianas, red represents trees. Confidence ellipses are represented for each life form (lianas and trees). For explanation of abbreviations, see Table 1.

**Figure 2 plants-11-03543-f002:**
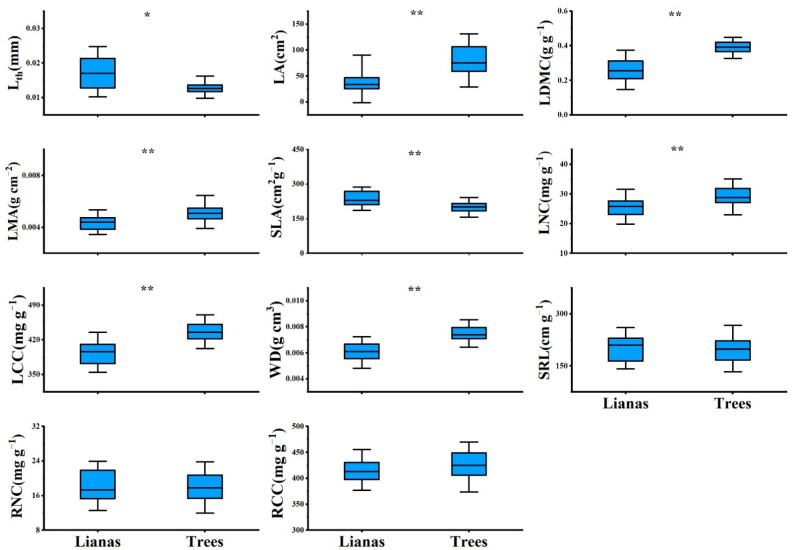
Leaf, stem, root traits of lianas and host trees based on 61 liana-host tree pairs in temperate forest of China. Asterisks indicate statistically significant difference between lianas and host trees at α < 0.05 (*) and α < 0.001 (**). For explanation of abbreviations, see Table 1.

**Figure 3 plants-11-03543-f003:**
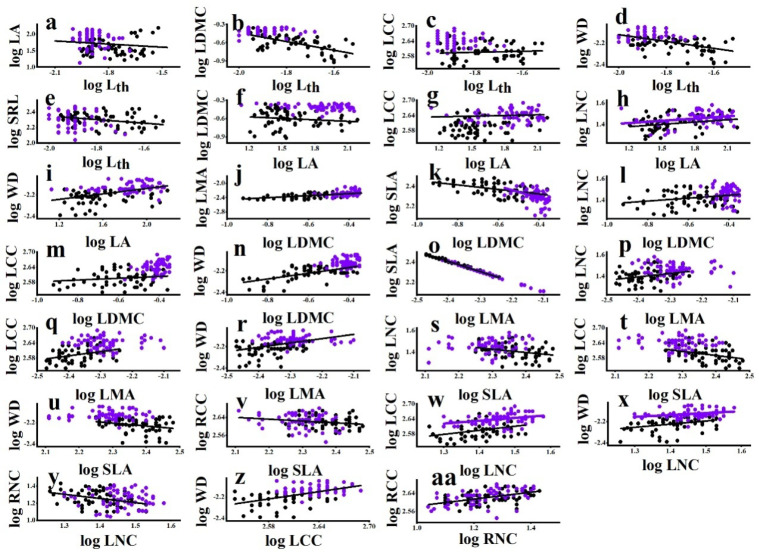
Bivariate trait relationships among liana and tree species from the temperate forest. Standardized Major Axis (SMA) coefficients for differences in slope and shifts in elevation between lifeforms are given in Appendix A. Filled black dots represent lianas and purple empty dots trees. Differences between lianas and trees are indicated by a black solid line and a purple solid line, respectively. The single solid line represents a significant association but no differences between lianas and trees. For explanation of abbreviations, see Table 1.

**Table 1 plants-11-03543-t001:** Functional traits measured in this study.

Traits	Abbreviation	Units
Leaf thickness	L_th_	mm
Leaf area	LA	cm^2^
Leaf dry matter content	LDMC	g g^−1^
Leaf mass per area	LMA	g cm^−2^
Specific leaf area	SLA	cm^2^ g^−1^
Leaf nitrogen content	LNC	mg g^−1^
Leaf carbon content	LCC	mg g^−1^
Wood density	WD	g cm^−3^
Specific root length	SRL	cm g^−1^
Root nitrogen content	RNC	mg g^−1^
Root carbon content	RCC	mg g^−1^

**Table 2 plants-11-03543-t002:** Principal component analysis results of functional traits for lianas and trees.

Functional Traits	PC1	PC2	PC3
L_th_	–0.28	0.11	–0.28
LA	0.24	−0.27	0.06
LDMC	0.43	0.07	0.19
LMA	0.36	0.34	–0.42
SLA	–0.38	−0.31	0.39
LNC	0.27	−0.28	0.17
LCC	0.38	−0.08	0.13
WD	0.41	−0.09	0.25
SRL	0.12	−0.17	−0.45
RNC	−0.05	0.53	0.35
RCC	0.03	0.53	0.33
% Total	36.5	16.5	10.4

For explanation of abbreviations, see Table 1.

**Table 3 plants-11-03543-t003:** Pearson correlation analysis of functional traits for lianas and trees.

	L_th_	LA	LDMC	LMA	SLA	LNC	LCC	WD	SRL	RNC
LA	−0.216 *									
LDMC	−0.6911 **	0.283 **								
LMA	−0.128	0.174	0.602 **							
SLA	−0.128	−0.172	−0.604 **	−0.997 **						
LNC	−0.147	0.417 **	0.295 *	0.224 *	−0.226 **					
LCC	−0.316 **	0.452**	0.554 **	0.452 *	−0.455 **	0.484 **				
WD	−0.514 **	0.478 **	0.734 **	0.434 **	−0.43 **	0.468 **	0.561 **			
SRL	−0.206 *	0.153	0.17	0.123	0.114	0.044	0.087	0.097		
RNC	0.087	−0.152	−0.067	0.07	−0.08	−0.198 *	−0.113	−0.138	−0.16	
RCC	0.032	−0.075	0.072	0.177	−0.182	0.118	0.016	0.04	−0.093	0.46 **

For explanation of abbreviations, see Table 1. Asterisks indicate statistically significant correlation between traits at α < 0.05 (*) and α < 0.001 (**).

## Data Availability

Data is available on request from the lead author.

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
