# Peer review of "Temperate Lianas Have More Acquisitive Strategies than Host Trees in Leaf and Stem Traits, but Not Root Traits"

_plants, 2022, doi:10.3390/plants11243543_

Round 1
Reviewer 1 Report
Ze Zhou et al. present a study where they analyze functional traits of leaves, stems and roots in trees and lianas in a temperate forest in China.
The work is interesting, presenting relevant data for understanding plant functional strategies and their coordination.
Global change includes all global changes. If they do not refer to climate change it would be more specific.
Line 26. How does it affect the ability to be a carbon sink?
Line 30. Improve the terms that are used, that talk about global change, or climate change.
Line 46. Improve wording, what they present is more acquisitive characteristics. Not acquisitive characteristics.
Line 48. It would be the other way around, if they have more acquisitive characteristics they would be less tolerant to drought! This is interesting in view of the results, since the no difference found in root traits could indicate that in the face of water limitation, the lianas would not be as sensitive.
Line 50. Check this, it is not like this! Lianas have lower WD values.
Include list of species analyzed
Include the full name of the characters in the legend of the figures.
Line 224. Which ecologist?
Line 245. Maybe better exploring rather than adapting.
Author Response
- Global change includes all global changes. If they do not refer to climate change it would be more specific.
Response (Lines 26): Thank you for your advice. Following the advice, we have specified the description.
- Line 26. How does it affect the ability to be a carbon sink?
Response: Thank you for your comments. Forests can act as a major terrestrial carbon sink by absorbing carbon dioxide. However, strength of this carbon sink varies both spatially and temporally as a result of forest composition and structure. Therefore, global change can affect the capacity of the forests as a carbon sink by altering the vegetation composition and structure of many forest ecosystems.
- Line 30. Improve the terms that are used, that talk about global change, or climate change.
Response (Lines 30): Thank you for your advice. Following the advice, we have specified the terms.
- Line 46. Improve wording, what they present is more acquisitive characteristics. Not acquisitive characteristics.
Response (Lines 45-47): Thank you for your advice. Following the advice, we have changed the wording.
- Line 48. It would be the other way around, if they have more acquisitive characteristics they would be less tolerant to drought! This is interesting in view of the results, since the no difference found in root traits could indicate that in the face of water limitation, the lianas would not be as sensitive.
Response (Lines 48-50): Thank you for your comments. We have rewritten the sentence to try to explain based on the previous studies. Dry or fragmented environments can provide higher resource conditions, such as high-light and nutrient-rich soils. Lianas with a more acquisitive strategy have more competitive ability for resource in this environment than trees. Of cause, as you said, more acquisitive strategy would be less tolerant to drought. However, some studies have found that plant economic spectrum is decoupled with water-use strategies. In the future, therefore, more studies should be conducted to test the difference between lianas and trees in water-use strategies, which may explain how lianas adapt to drought environment.
- Line 50. Check this, it is not like this! Lianas have lower WD values.
Response: Thank you for your advice. We have checked, and lianas indeed have no significant difference in wood density in the study.
- Include list of species analyzed
Response: Thank you for your advice. We have included the list of species in the Table S1.
- Line 224. Which ecologist?
Response (lines 233-234): Not one ecologist proposed the leaf economics spectrum. We have rewritten the sentence more properly.
- Line 245. Maybe better exploring rather than adapting.
Response (line 254): Thank you for your advice. We have corrected the wording following your advice.
Reviewer 2 Report
This is an interesting study addressing a relevant topic and based on large data set of field-measured plant traits and appropriate analyses. The paper explores leaf, stem and root traits of lianas and host trees in a temperate forest. They found more acquisitive leaf and stem traits in lianas compared to host trees while root traits were not correlated with aboveground plant traits. These results are novel for temperate forests and consistent with the findings described in tropical forests. The text is logical and well written. I have only few comments that could help to further increase the impact of this paper.
Abstract
It is well-written and informative.
Introduction
Introduction provides interesting and appropriate research background and it has been written with clear structure and logic. Novelties are explicitly defined, hypotheses are relevant and well-explained. I only missed some details of existing field evidences about correlations between above and belowground traits (L 53-57). Authors cited Reich and Cornelissen (2014) paper. However, it was not clear if this paper described only a theory (expected patterns) or it was a synthesis of the related field evidences.
Material and Methods
L. 85-88 Sampling was conducted in a Quercus forest where climbing plants were abundant. Please, add information about land use history (disturbance history), about landscape context (fragmentation) or other factors that could explain the high abundance of climbing plants in this vegetation type. Are these forests representative for the typical temperate forests with climbing plants? (cf. Allen et al. 2007 about floodplain forests as typical temperate forests with many lianas L 67)
L 87-89 Please, add the time period (for example 1990-2020) represented by these mean temperature and precipitation meteorological data.
L 90-94 “line transects with 100 m altitudinal intervals”How many transects were sampled? Please add a figure in appendix showing the detailed study design.
Trait sampling: Please, add information about the season when plant traits were sampled.
Please, indicate whether you used the original trait data in analyses or you applied some data transformations. For example, bivariate trait relationships were calculated at log-log scales (Fig.3.). However, it is not clear what transformations were used in PCA (Fig.1) or for Pearson correlations (Table 3).
According to the sampling design (L 90-94) 61 liana-host tree pairs were sampled, i.e. liana and tree data were not independent. How did you consider this specific data structure in analyses? You described data structure of GLM but without mentioning these relationships (L 130-132).
Results
Clearly presented, consistent with hypotheses giving convincing responses to the study questions.
Discussion and Conclusions
I enjoyed reading these texts. Discussion is well-written, clearly structured and it explains properly the results. I agree with the comment of authors about the urgent need of long-term observations about the temporal dynamics of lianas in temperate ecosystems. I also liked the argumentation explaining the lack of correlations between above-ground traits and root traits in these communities (L 235-249). Authors state that these traits are “not always” coordinated (L 16) and they explain how they are decoupled (L 235-249). However, in this way they also suggest implicitly that the basic state should be the coordinated version (cf. whole plant concept, trade offs, allocation principles). I missed here a short description of this coordinated reference state and the related mechanisms.
In this paper, the relationships across life forms (across trees and lianas) have been explored. As an outlook (perspective) for future studies, it would be interesting to add a short discussion about the potential correlations and mechanisms within life form and within organisms. The data set analyzed here would be appropriate also for the separate analyses of life forms exploring within life form relationships. Conclusions are well-supported by data and results.
Supplementary Materials
I could download this material. However, the related text:“The following are available online at www.mdpi.com/xxx/s1 “ was not a valid address (L 314). Please, provide it during the revision.
Author Response
- Introduction: Introduction provides interesting and appropriate research background and it has been written with clear structure and logic. Novelties are explicitly defined, hypotheses are relevant and well-explained. I only missed some details of existing field evidences about correlations between above and belowground traits (L 53-57). Authors cited Reich and Cornelissen (2014) paper. However, it was not clear if this paper described only a theory (expected patterns) or it was a synthesis of the related field evidences.
Response (lines 58-60): Thank you for your comments. This paper described an expected pattern based on the related theory and few field evidences. We added a field study to provide some details of existing field evidences about relationship between above and belowground traits in self-supporting plants.
- Material and Methods:
L85-88 Sampling was conducted in a Quercus forest where climbing plants were abundant. Please, add information about land use history (disturbance history), about landscape context (fragmentation) or other factors that could explain the high abundance of climbing plants in this vegetation type. Are these forests representative for the typical temperate forests with climbing plants? (cf. Allen et al. 2007 about floodplain forests as typical temperate forests with many lianas L 67)
Response (lines 90-93): Thank you for your advice. Following the advice, we have described the disturbance history of the sampled area.
L87-89 Please, add the time period (for example 1990-2020) represented by these mean temperature and precipitation meteorological data.
Response (line 95): Thank you for your advice. Following the advice, we have added the time period represented by the meteorological data.
L90-94 “line transects with 100 m altitudinal intervals” How many transects were sampled?
Response (lines 100-101): Thank you for your advice. Following the advice, we have added the information on the transects. In our studies, a total of ten transects were set to investigated the lianas and host trees.
Trait sampling: Please, add information about the season when plant traits were sampled.
Response (lines 100-101): Thank you for your advice. Following the advice, we have added the information about the time of sampling.
Please, indicate whether you used the original trait data in analyses or you applied some data transformations. For example, bivariate trait relationships were calculated at log-log scales (Fig.3.). However, it is not clear what transformations were used in PCA (Fig.1) or for Pearson correlations (Table 3).
Response (lines 143-144): Thank you for your advice. For PCA, the raw traits data was standardized by transforming the normalized trait values into z-scores before analyzing. For the pearson correlation, we also standardized each functional trait and used the standard Pearson correlation analyses. We have added the information.
According to the sampling design (L 90-94) 61 liana-host tree pairs were sampled, i.e. liana and tree data were not independent. How did you consider this specific data structure in analyses? You described data structure of GLM but without mentioning these relationships (L130-132).
Response: Thank you for your comments. In our study, in order to decrease the effect of non- independent data we used generalized linear mixed effects models to evaluate the difference between lianas and trees by using family was a random effect.
In this paper, the relationships across life forms (across trees and lianas) have been explored. As an outlook (perspective) for future studies, it would be interesting to add a short discussion about the potential correlations and mechanisms within life form and within organisms. The data set analyzed here would be appropriate also for the separate analyses of life forms exploring within life form relationships. Conclusions are well-supported by data and results.
Response (lines 313-322): Thank you for your nice advices. Indeed, exploring the difference and relationships within life form are also useful to understand the potential mechanisms. We have added a short discussion about this topic in the end of the discussion section. We will do this work in the next study by collecting more data.